# Association Between Dynapenia/Sarcopenia, Extrapyramidal Symptoms, Negative Symptoms, Body Composition, and Nutritional Status in Patients with Chronic Schizophrenia

**DOI:** 10.3390/healthcare13010048

**Published:** 2024-12-30

**Authors:** Reiko Kamoi, Yoshihiro Mifune, Krishan Soriano, Ryuichi Tanioka, Risa Yamanaka, Hirokazu Ito, Kyoko Osaka, Hidehiro Umehara, Rie Shimomoto, Leah Anne Bollos, Rick Yiu Cho Kwan, Itsuro Endo, Sr. Sahlee Palijo, Katsuhiro Noguchi, Kazushi Mifune, Tetsuya Tanioka

**Affiliations:** 1Mifune Hospital, Kagawa 763-0073, Japan; reihamlintel@gmail.com (R.K.); ymifune29@gmail.com (Y.M.); mifune@mifune-hp.jp (K.M.); 2Graduate School of Health Sciences, Tokushima University, Tokushima 770-8509, Japan; ksoriano@spup.edu.ph (K.S.); yamanaka0025@gmail.com (R.Y.); leahaclb@gmail.com (L.A.B.); 3Graduate School, St. Paul University Philippines, Tuguegarao City 3500, Cagayan, Philippines; sahlee011172@gmail.com; 4Faculty of Health Sciences, Hiroshima Cosmopolitan University, Hiroshima 731-3166, Japan; tanioka@hcu.ac.jp; 5Graduate School of Biomedical Sciences, Tokushima University, Tokushima 770-8509, Japan; h.itoh@tokushima-u.ac.jp (H.I.); sanntyoumenoumehara@gmail.com (H.U.); endoits@tokushima-u.ac.jp (I.E.); 6Department of Nursing, Nursing Course of Kochi Medical School, Kochi University, Kochi 783-8505, Japan; osaka@kochi-u.ac.jp (K.O.); shimomot@kochi-u.ac.jp (R.S.); 7School of Nursing, Tung Wah College, Hong Kong, China; rickkwan@twc.edu.hk; 8Department of Neuropsychiatry, Faculty of Medicine, Kagawa University, Kagawa 760-8521, Japan; noguchi.katsuhiro@kagawa-u.ac.jp

**Keywords:** dynapenia, sarcopenia, schizophrenia, extrapyramidal symptom, nutritional status

## Abstract

**Background/Objectives:** This study aimed to determine the association between chronic schizophrenia, extrapyramidal symptoms (EPSs), body composition, nutritional status, and dynapenia/sarcopenia. **Methods:** Data from 68 chronic patients with schizophrenia were analyzed using Spearman’s rho correlation coefficients, Kruskal–Wallis test, Mann–Whitney U test, and Cramér’s V statistics. **Results:** Among the participants, 32.4% had no loss of muscle mass or function, 39.7% had dynapenia, and 27.9% had sarcopenia. This study identified five key findings: (1) Bilateral grip strength, skeletal muscle index, and walking speed are interrelated, with higher negative symptom scores linked to slower movement and rigidity, particularly in the sarcopenia group, indicating that negative symptoms may contribute to muscle weakness and progression to sarcopenia. (2) Increasing age is associated with a decrease in chlorpromazine equivalent dose and an increase in the severity of EPSs. (3) Blood urea nitrogen (BUN)/creatinine ratio and all sarcopenia risk indicators were significantly negatively correlated. (4) Dynapenia and sarcopenia groups exhibited significant differences in muscle mass and nutritional status compared to the non-penia group, including reduced muscle mass, lower basal metabolic rate, and lower visceral fat levels. (5) There was an association between the Barthel Index (BI) score for activities of daily living (ADL) and dynapenia/sarcopenia. Particularly with regard to ADL, it seems necessary to pay attention to muscle weakness in partially independent patients who score 60 points or more. **Conclusions:** BUN/creatinine ratio, BI, EPSs, body mass index, grip strength, total protein, and albumin were useful indicators for detecting the risk of dynapenia/sarcopenia in routine psychiatric care.

## 1. Introduction

Schizophrenia is one of the world’s most disabling mental disorders [1]. Negative symptoms significantly contribute to poor quality of life due to impairments in everyday functioning among patients [2]. Social withdrawal and lack of motivation contribute to unhealthy habits and a sedentary lifestyle, which Tanioka et al. have linked to an increased risk of muscle weakness, metabolic diseases, and higher mortality rates [3]. Research by Bulbul et al. [4] has revealed that sarcopenia—characterized by decreased muscle mass, reduced muscle strength or dynapenia, and impaired physical function [5]—is more prevalent among patients with schizophrenia than in the general population, largely due to these factors.

Antipsychotic medication affects metabolic indices and carries the risk of medication-induced extrapyramidal symptoms (EPSs), such as tardive dyskinesia, dystonia, and particularly parkinsonism [6]. EPSs, characterized by tremors, rigidity, and slowed motor function in the truncal region and extremities, have been associated with gait imbalance, falls, and reduced physical activity in patients with schizophrenia [7].

Nutritional status is a critical factor for the health of patients with schizophrenia. Low body weight is associated with a high mortality rate, whereas obesity increases the risk of premature death [8]. Malnutrition may occur even in overweight individuals [9,10], and the prevalence of undernutrition and being underweight is high in this population [11]. It has been reported that weak grip strength has a high sensitivity for detecting the presence or likelihood of malnutrition, further highlighting the vulnerability of these patients to muscle wasting and nutritional deficiencies [12]. Falls and fractures are significant health concerns among patients with schizophrenia due to their association with diminished quality of life, impairment, and increased mortality. Patients with an earlier onset of schizophrenia have a higher incidence of hip fractures, which requires bone health management and fracture prevention measures [13].

Clinical assessment tools, such as the negative symptoms scale, are crucial for evaluating vulnerability risk by considering a wide range of mental and physical health factors, including frailty, quality of life, physical activity, nutritional status, and polypharmacy [14]. Increased biomarkers of oxidative stress and inflammation in schizophrenia have been linked to the accelerated aging hypothesis, with growing evidence suggesting that oxidative stress contributes to the development of frailty in these patients [15,16]. Additionally, a high blood urea nitrogen to creatinine (BUN/creatinine) ratio has been associated with an increased risk of sarcopenia, particularly in relation to skeletal muscle index (SMI) [17] and grip strength [18], and may serve as a predictor of sarcopenia or frailty in chronically hospitalized patients with schizophrenia.

Sarcopenia has a significant impact on daily life, affecting basic movements such as walking and standing up, making it easier to fall over [4,19]. Clarifying the interrelation between risk indicators in patients with schizophrenia will help address these invalidating features.

The aim of this study was to determine the association between dynapenia/sarcopenia, extrapyramidal symptoms, negative symptoms, body composition, and nutritional status in patients with schizophrenia.

Specifically, we extracted four problem statements to address the main research question: (1) How do muscle mass and function vary with EPSs? (2) What is the correlation between sarcopenia risk indicators and muscle mass, muscle function, and body composition? (3) What are the differences in muscle mass and function between groups without dynapenia/sarcopenia and groups with dynapenia or sarcopenia? (4) What are the associations between the Barthel Index score for activities of daily living, clinical assessment of dynapenia/sarcopenia, and laboratory risk indicators?

## 2. Materials and Methods

### 2.1. Study Design

This study used the observational cross-sectional study design. Therefore, we followed the Strengthening the Reporting of Observational Studies in Epidemiology (STROBE) checklist to report the methods and findings of this study [20]. The data were collected from 18 May 2024 to 30 September 2024.

### 2.2. Setting

Mifune Hospital in Japan was founded in 1953, and it is a 328-bed psychiatric hospital. It has departments of psychiatry, internal medicine, dentistry, and oral surgery.

### 2.3. Participants

Participants who fulfilled the eligibility criteria below were recruited into this study.

#### 2.3.1. Inclusion Criteria

All inpatients who were diagnosed with schizophrenia according to the criteria in the fifth edition of the Diagnostic and Statistical Manual of Mental Disorders (DSM-5), were aged 18 years or above, did not meet any of the exclusion criteria, and were able to provide informed consent were enrolled in this study.

#### 2.3.2. Exclusion Criteria

Patients were excluded who had severe mental disorders other than schizophrenia, as defined by DSM-5 criteria (e.g., delusional disorders, bipolar and related disorders, depressive disorders, neurodevelopmental disorders, substance-related and addictive disorders, or personality disorders), were unable to understand instructions due to a medical condition or medication status, or were advised against participating in the study by their attending psychiatrists.

Additionally, patients were not included if they had physical diseases (e.g., cerebrovascular disease, stroke or other neurological diseases, cancer, dehydration, reduced blood flow to the kidneys, congestive heart failure, gastrointestinal bleeding, urinary tract obstruction, acute kidney injury, chronic kidney disease) or carried pacemakers, implantable cardioverter defibrillators, and cardiac resynchronization therapy-defibrillators.

### 2.4. Variables

Variables are classified into four areas:Demographics: age and gender.Blood parameters: alanine aminotransferase (ALT), aspartate aminotransferase (AST), albumin, total cholesterol (TC), triglycerides (TG), gamma-glutamyl transferase (γ-GT), high-density lipoprotein cholesterol (HDL-C), low-density lipoprotein cholesterol (LDL-C), blood urea nitrogen (BUN), creatinine, BUN/creatinine ratio, blood sugar level, uric acid (UA), and hemoglobin A1C (HbA1C).Clinical assessment: EPSs, names and doses of atypical antipsychotics, global functioning, negative symptoms, ability to perform activities of daily living (ADL).Muscle mass and function, as well as body composition.

### 2.5. Measurement

#### 2.5.1. Demographics

Age and gender were retrieved from the medical records of the participants.

#### 2.5.2. Blood Parameters

For ALT, AST, albumin, TC, TG, γ-GT, HDL-C, LDL-C, BUN, creatinine, and UA, blood samples of 4.5 mL were collected in tubes containing a coagulation accelerator and separator and analyzed using the “Accute RX TBA-400FR” discrete clinical chemistry autoanalyzer manufactured by Canon Medical Systems Corporatio, Tochigi, Japan.

For blood glucose, the blood sample was collected in a 2 mL tube containing EDTA-2K + NaF and analyzed using the same system.

For HbA1C, the blood sample was collected in a 2 mL tube containing EDTA-2K and analyzed using the “A1c GEAR S” manufactured by Sakae Co., Tokyo, Japan.

#### 2.5.3. Clinical Assessment

Twelve trained psychiatrists assessed all of the enrolled subjects utilizing the following scales. All raters had at least 9 (Median 25, range: 9–49) years of experience administering psychiatric rating scales to people with schizophrenia. To minimize bias, the assessors were blinded to the participant’s classification.

Negative symptoms were measured using the Brief Negative Symptoms Scale (BNSS) [21]. The BNSS is a hetero-evaluation tool consisting of 13 items that assess five domains: anhedonia, asociality, avolition, blunted affect, and alogia. Each item is rated on a scale from 0 to 6, with higher scores indicating greater symptom severity. Two primary clinical factors were utilized: the motivation and pleasure factor (sum of items 1–3 and 5–8) and the expression factor (sum of items 9–13). Rater training involved a thorough review of the BNSS manual and workbook, as well as reading a series of training manuals provided by the original author of the BNSS. Specifically, the first author of this study (R.K.) carefully read and viewed the main manuals and associated training manuals. As a senior psychiatrist with experience using assessment tools, the first author provided a tutorial for the raters.

EPSs were measured using the Drug-Induced Extrapyramidal Symptoms Scale (DIEPSS) [22]. This practical, multidimensional rating scale is designed to assess drug-induced extrapyramidal side effects and quantify the severity of objectively observed symptoms. The DIEPSS consists of 8 individual items, each rated by objective observation on a 5-point scale, where 0 indicates normal, and 4 indicates the most severe. The items include gait, bradykinesia, sialorrhea, rigidity, tremor, akathisia, dystonia, and dyskinesia.

The types and doses of atypical antipsychotics were also recorded. The types of medications used in these patients included olanzapine, risperidone, aripiprazole, ziprasidone, clozapine, amisulpride, quetiapine fumarate, and paliperidone extended-release tablets. The daily doses of the prescribed antipsychotic drugs were converted to the equivalent daily dose of chlorpromazine per 100 mg, based on the international consensus [23,24,25].

Global functioning was measured using the Global Assessment of Functioning (GAF) scale [26], which evaluates the psychological, social, and occupational levels of functioning among the subjects. The scale is administered by a clinician and provides a score between 1 and 100 for the current or previous period, with higher scores indicating better functioning.

The ability to perform ADL was assessed using the Barthel Index (BI) [27].

Seventeen trained registered nurses (including head nurses and chief nurses) assessed all of the enrolled subjects. The BI is an ordinal scale that includes 10 important questions regarding performance in daily activities. Higher scores indicate greater independence in performing these activities. Based on previous research [28], the participants were classified as severe (0–20), moderate (21–60), mild (61–90), and with no disability (91–100).

#### 2.5.4. Muscle Mass and Function

Muscle mass and function were classified into three categories according to the Asian Working Group for Sarcopenia (AWGS) [29] guideline: normal (without dynapenia/sarcopenia), dynapenia, and sarcopenia. The “without dynapenia/sarcopenia” classification is defined by having muscle mass, strength, and physical performance within the typical range for healthy individuals. “Dynapenia” is characterized by muscle weakness without a significant loss of muscle mass. “Sarcopenia”, on the other hand, involves both low muscle mass and muscle weakness or impaired physical performance.

Body composition was measured using the RD-545 InnerScan Pro (TANITA Corporation, Tokyo, Japan), which included body weight, body fat, muscle mass, the weight of bone minerals in the body, visceral fat level, basal metabolic rate, metabolic age, total body water, and body mass index (BMI) [30]. This device can estimate fat and muscle analysis individually for arm, leg, and trunk segments (i.e., left arm, right arm, left leg, right leg, and trunk) [31].

The total limb skeletal muscle index (SMI) (kg) was calculated from the information obtained from the body mass, and the data were divided by the square of the corresponding height (m^2^). Significant loss of muscle mass was defined as total limb SMI less than 7.0 kg/m^2^ for men and less than 5.7 kg/m^2^ for women [29].

The following are procedural details for body composition analysis.

Participants were asked to refrain from fluid intake (including water) for 2 h prior to measurements. No alcohol was allowed the day before the measurements, and measurements were taken at least 2 h after awakening, eating, or bathing.

Measurement time of day: around 10 a.m. or 2 p.m.

Measurement Conditions: The manufacturer’s instructions were followed to avoid conditions that could interfere with the correct measurements, such as fever, diarrhea, or other physical symptoms; dehydration or swelling; exposure to cold air for a long time; or hypothermia. It was recommended that measurements be taken with bare feet and an empty bladder and bowel. The indications for a correct posture were also followed, instructing the patients not to touch the skin on their armpits or inner thighs, not to bend their elbows or knees, and to avoid moving or talking while taking measurements. Good electrode contact was ensured with the palms and soles of hands and feet, which were cleaned and slightly moisturized if dry. The measurements were taken on a hard, flat, and stable floor, with the patient as close to naked as possible.

Muscle strength was measured by using a digital grip dynamometer (T.K.K.5401; Takei Scientific Instruments, Co., Ltd., Niigata, Japan). Grip strength was used as an indicator of muscle weakness and was defined as a grip strength of 26 and 18 kg for men and women, respectively [29].

Using a gait analysis system (manufactured by NEC Corporation, Tokyo, Japan) [32], we measured the walking speed of patients walking at a normal pace over a distance of approximately 6 m.

### 2.6. Statistical Methods

Data were expressed as median and interquartile range for non-normal distributions. Categorical data were expressed as numbers and percentages (Table 1). Spearman’s rho (ρ) correlation coefficients were calculated to relate the variables (Table 2 and Table 3). Multigroup comparisons, without dynapenia/sarcopenia, dynapenia, and sarcopenia groups were performed using the nonparametric Kruskal–Wallis test with the Dwass–Steel–Critchlow–Fligner (DSCF) procedure as a post-hoc analysis (Table 4). The Mann–Whitney U test was used to compare differences between two specific groups with and without dynapenia/sarcopenia (Figure 1). Cross tabulation with Cramér’s V was used to reveal the association between BI level, dynapenia/sarcopenia, and BUN/creatinine value (Table 5). All statistical analyses were performed using the Jamovi Statistical Software Version 2.4.11.0 (The Jamovi Project, Sydney, Australia) [33]. Statistical significance was set at *p* < 0.05.

Sample size calculation was performed using G*Power software (ver. 3.1.9.7;) [34].

Assuming a correlation bivariate normal model with a two-tailed test, an assumed moderate correlation (H1 = 0.34), an alpha level of 0.05, and a power of 0.80, the required sample size was calculated to be 65 cases.

Assuming a one-way analysis of variance between the three groups with an effect size of 0.40, alpha level of 0.05, and power of 0.80, the required sample size was calculated to be 66 cases.

Assuming two independent groups with an effect size = 0.80, alpha level of 0.05, and statistical power = 0.80, the required sample size was calculated to be 58 cases.

## 3. Results

### 3.1. Number of Participants Who Completed All of the Tests

A total of 173 patients were enrolled, consented, and participated in the study, but only 68 had complete sets of test results available. The remaining 105 patients were excluded from the analyses because they were unable to undergo the measurement of walking speed or body composition (N = 91) and/or to undergo blood tests (N = 92) owing to severe mental symptoms such as delusions.

### 3.2. Descriptive Data

Descriptive data of all variables are shown in Table 1. The median (min–max) age of the participants is 64 years (28–86); height is 160 cm (139–177); weight is 56.5 kg (31.0–94.9); BMI is 22.35 (13.4–34.1); GAF is 25 (5–35); BI is 95 (15–100); DIEPPS overall severity level is 1 (0–3); chlorpromazine equivalent dose is 629 mg (range: 12.6–2382).

The study sample was evenly distributed, with 34 males (50%) and 34 females (50%). Of the participants, 22 (32.4%) exhibited no loss of muscle mass or function (without dynapenia/sarcopenia), whereas 27 (39.7%) experienced dynapenia, characterized by a reduction in muscle strength without corresponding muscle mass loss. There were 19 patients with sarcopenia (27.9%), which indicates a loss of muscle mass and strength. DIEPSS overall severity revealed that 21 (30.9%) participants had a severity score of 0, indicating no observable EPSs, whereas 18 (26.5%) participants received a severity score of 1, indicating very mild or uncertain symptoms. In addition, 26 participants (38.2%) received a severity score of 2, indicating mild symptoms, and 3 participants (4.4%) received a severity score of 3, indicating moderate symptoms. There were no participants with a rating of 4.

### 3.3. Main Results

#### 3.3.1. Problem Statement #1: Correlations Between Muscle Mass and Function and EPSs

As shown in Table 2, a strong positive correlation was observed between left and right grip strength (ρ = 0.9, *p* < 0.001). Both left and right grip strength were also significantly correlated with SMI (ρ = 0.7, *p* < 0.001 for left grip; ρ = 0.7, *p* < 0.001 for right grip). Walking speed was positively correlated with both left grip strength (ρ = 0.3, *p* < 0.05) and right grip strength (ρ = 0.3, *p* < 0.05), as well as with SMI (ρ = 0.3, *p* < 0.05). There was a significant correlation between sarcopenia and EPSs as measured by the DIEPSS. Specifically, gait disturbance (DIEPSS 1) was negatively correlated with left grip strength (ρ = −0.4, *p* < 0.01), right grip strength (ρ = −0.4, *p* < 0.01), and SMI (ρ = −0.3, *p* < 0.05). Bradykinesia (DIEPSS 2) showed a significant negative correlation with grip strength (left: ρ = −0.24, *p* < 0.05; right: ρ = −0.3, *p* < 0.05) and walking speed (ρ = −0.4, *p* < 0.001). It was also significantly correlated with overall DIEPSS severity (DIEPSS 9) (ρ = 0.7, *p* < 0.001). Muscle rigidity (DIEPSS 4) correlated negatively with the chlorpromazine equivalent dose (ρ = −0.5, *p* < 0.001), and tremor (DIEPSS 5) correlated with gait disturbance (DIEPSS 1) (ρ = 0.3, *p* < 0.01) and bradykinesia (DIEPSS 2) (ρ = 0.3, *p* < 0.05).

The total Brief Negative Symptom Scale (BNSS) score showed a positive correlation with the total DIEPSS severity (DIEPSS 9) (ρ = 0.2, *p* < 0.05). The BNSS score was particularly related to bradykinesia (DIEPSS 2) (ρ = 0.4, *p* < 0.001). Muscle rigidity (DIEPSS 4) also correlated with the BNSS total score (ρ = 0.3, *p* < 0.01). Examining the individual subscales of the BNSS, motivational deficit (F1) correlated with bradykinesia (DIEPSS 2) (ρ = 0.4, *p* < 0.01) and muscle rigidity (DIEPSS 4) (ρ = 0.3, *p* < 0.05), while expressive deficit (F2) correlated with bradykinesia (DIEPSS 2) (ρ = 0.4, *p* < 0.01) and muscle rigidity (DIEPSS 4) (ρ = 0.3, *p* < 0.01).

There were also significant correlations between age and chlorpromazine equivalent dose (ρ = −0.5, *p* < 0.001), as well as age and the severity of EPSs (ρ = 0.3, *p* < 0.05). Additionally, the severity of EPSs was negatively correlated with the chlorpromazine equivalent dose (ρ = −0.4, *p* < 0.001).

#### 3.3.2. Problem Statement #2: Correlations Between BUN/Creatinine Ratio and Muscle Mass, and Function, and Body Composition

As shown in Table 3, BUN/creatinine ratio showed significant negative correlations with BMI (ρ = −0.5, *p* < 0.001), body fat percentage (ρ = −0.3, *p* < 0.05), visceral fat level (ρ = −0.5, *p* < 0.001), basal metabolic rate (ρ = −0.5, *p* < 0.001), bone mass (ρ = −0.5, *p* < 0.001), left grip strength (ρ = −0.3, *p* < 0.05), right grip strength (ρ = −0.3, *p* < 0.05), SMI (ρ = −0.4, *p* < 0.001), and walking speed (ρ = −0.3, *p* < 0.05).

Age showed significant negative correlations with several body composition metrics, including BMI (ρ = −0.5, *p* < 0.001), body fat percentage (ρ = −0.4, *p* < 0.001), visceral fat level (ρ = −0.4, *p* < 0.001), basal metabolic rate (ρ = −0.5, *p* < 0.001), and bone mass (ρ = −0.4, *p* < 0.001).

Body water percentage showed negative correlations between BMI (ρ = −0.77, *p* < 0.001), body fat percentage (ρ = −0.89, *p* < 0.001), visceral fat level (ρ = −0.7, *p* < 0.001), and basal metabolic rate (ρ = −0.3, *p* < 0.05).

#### 3.3.3. Problem Statement #3: Comparison of Variables Between Groups Defined by Muscle Mass and Function: Without Dynapenia/Sarcopenia vs. Dynapenia vs. Sarcopenia

As shown in Table 4, variables were compared across the groups categorized by muscle mass and function: without dynapenia/sarcopenia, dynapenia, and sarcopenia. The total BNSS score was significantly higher in the sarcopenia group compared to the dynapenia group (*p* < 0.05). Significant differences in age were found between the groups, with the sarcopenia group being the oldest and the without dynapenia/sarcopenia group being the youngest. The comparison showed a significant difference in age between the sarcopenia group (*p* < 0.01) and the dynapenia group (*p* < 0.05) compared to the without dynapenia/sarcopenia group.

All measures of muscle mass, including left and right arms (*p* < 0.01), left and right legs (*p* < 0.05 to *p* < 0.001), trunk muscle mass (*p* < 0.05 to *p* < 0.001), and SMI (*p* < 0.05 to *p* < 0.001) showed significant differences among the groups. The sarcopenia group had significantly lower muscle mass compared to both the dynapenia and without dynapenia/sarcopenia groups. Additionally, the sarcopenia group had a significantly lower BMI than both the dynapenia and without dynapenia/sarcopenia groups (*p* < 0.001). The sarcopenia group also exhibited a lower basal metabolic rate, indicating reduced energy expenditure compared to the dynapenia and without dynapenia/sarcopenia groups (*p* < 0.05 to *p* < 0.001). Bone mass was significantly lower in the sarcopenia group compared to both the without dynapenia/sarcopenia and dynapenia groups (*p* < 0.001).

However, there were no significant differences among the groups for total DIEPSS score, chlorpromazine equivalent dose, body water, AST, γ-GT, AG ratio, T-Cho, HDL-C, TG, HbA1c, creatinine, body fat, and UA levels.

#### 3.3.4. Problem Statement #4: Association Between BI Level, Dynapenia/Sarcopenia, and BUN/Creatinine Ratio

As indicated in Table 4, the BUN/creatinine ratio demonstrated a significant trend, with a *p*-value of 0.051. 

To further investigate this finding, a comparison was made between the two specific groups with and without dynapenia/sarcopenia, as illustrated in Figure 1. The Mann–Whitney U test revealed a significantly higher BUN/creatinine ratio in the group with dynapenia/sarcopenia compared to the group without dynapenia/sarcopenia (*p* = 0.03).

There was an association between BI score and dynapenia/sarcopenia (Cramer’s V = 0.41). Among patients with BI level 1 (no disability), none had a BUN/creatinine ratio of 10 or greater. All patients with BI levels 2–4 had a BUN/creatinine ratio of 10 or greater. Patients with BI levels 1–2 were also found to have dynapenia/sarcopenia (Table 5).

## 4. Discussion

### 4.1. Key Results

This study identified five key findings. First, bilateral grip strength, SMI, and walking speed are interrelated, with higher negative symptom scores linked to slower movement and rigidity, particularly in the sarcopenia group, indicating that negative symptoms may contribute to muscle weakness and progression to sarcopenia. Second, increasing age is associated with a decrease in chlorpromazine equivalent dose and an increase in the severity of EPSs. Third, the BUN/creatinine ratio and all sarcopenia risk indicators were significantly negatively correlated. Fourth, the dynapenia and sarcopenia groups exhibited significant differences in muscle mass and nutritional status compared to the non-penia group, including reduced muscle mass, lower basal metabolic rate, and lower visceral fat levels. Fifth, there was an association between BI score and dynapenia/sarcopenia. Particularly with regard to ADL, it seems necessary to pay attention to muscle weakness in partially independent patients who score 60 points or more.

### 4.2. Interpretation and Generalizability

The current findings reinforce the connection between EPSs, negative symptoms, and sarcopenia in chronic schizophrenia. Walking impairment (DIEPSS 1) negatively correlated with bilateral grip strength, SMI, walking speed, and chlorpromazine equivalent dose, indicating that muscle weakness and drug-induced pyramidal symptoms contribute to impaired mobility [35,36]. Slowness of movement (DIEPSS 2) exhibited a stronger correlation with decreased bilateral grip strength, decreased walking speed, and the chlorpromazine equivalent dose. Muscle rigidity (DIEPSS 4) also correlates with the chlorpromazine equivalent dose. Tremor (DIEPSS 5) was associated with walking speed, suggesting that tremor affects gait and contributes to decreased walking speed [36]. Grip strength has been proposed as a biomarker for overall health status [37], and our results align with this by showing that patients with EPSs and negative symptoms experience declines in mobility and muscle strength. Furthermore, DIEPSS overall severity (DIEPSS 9) is related to left and right-side muscle strength and SMI walking speed, suggesting that DIEPSS could be a helpful tool for estimating declines in muscle strength, walking speed (lower limb muscle strength), and SMI (Table 6).

A previous study has identified negative symptoms and a tendency toward sedentary behaviors as factors contributing to sarcopenia [38]. However, this study could not show a direct link between these factors and muscle weakness, decreased walking speed (lower limb muscle strength), and SMI.

Negative symptoms, particularly the lack of motivation and expressive deficits, were associated with higher total negative symptom scores, slowness of movement (DIEPSS 2), and muscle contracture (DIEPSS 4), highlighting the link between negative symptoms and EPSs. The results in Table 3 indicate that with increasing age, the chlorpromazine equivalent dose decreases while the severity of EPSs increases. These findings suggest that as patients age, they are increasingly likely to experience EPSs, prompting psychiatrists to prescribe lower doses of antipsychotic medications to minimize side effects. Additionally, the BUN/creatinine ratio and all sarcopenia risk indicators showed significant negative correlations, suggesting that the BUN/creatinine ratio may effectively confirm the risk of dynapenia and sarcopenia.

In Table 4, the dynapenia and sarcopenia groups displayed significant differences in muscle mass and nutritional status compared to the non-penia group. Specifically, both the dynapenia and sarcopenia groups exhibited decreased muscle mass, a lower basal metabolic rate, and reduced visceral fat levels. These findings highlight the necessity of addressing both psychiatric and physical health in schizophrenia, as muscle dysfunction and inadequate nutritional status directly affect patients’ overall health and may lead to additional complications, such as pneumonia [47].

The dynapenia and sarcopenia groups were significantly older than the non-dynapenia group, with minimum ages of 35 and 43 years, respectively. Therefore, it is crucial to acknowledge the prevalence of dynapenia and sarcopenia in patients with chronic schizophrenia, particularly in those younger than 50 years, making it essential for psychiatrists and nurses to be aware of this and consider appropriate care interventions. Muscle strength in both the upper and lower limbs was significantly greater in the non-penia group compared to the dynapenia and sarcopenia groups. It is well known that aging reduces muscle mass and is associated with sarcopenia [39,40]. Maintaining adequate muscle mass and strength is important.

Interestingly, the BNSS scores were significantly higher in the sarcopenia group than in the dynapenia group. Negative symptoms may influence the motivational factors of patients with schizophrenia [48], and in the sarcopenia group, they may also impact the activities and dietary intake necessary to maintain muscle mass [49]. Considering negative symptoms, increasing the amount of voluntary activity, rather than solely focusing on direct muscle training, may be crucial in preventing sarcopenia in patients with schizophrenia during the chronic phase [50,51].

As previously noted, the BUN/creatinine ratio and all sarcopenia risk indicators displayed significant negative correlations, suggesting that the BUN/creatinine ratio may be an effective indicator for assessing the risk of dynapenia and sarcopenia, as shown in Table 3. The analysis results presented in Table 4 revealed a significant trend regarding the BUN/creatinine ratio. A comparison was made between two specific groups—those with and without dynapenia/sarcopenia—as illustrated in Figure 1. The findings indicated a significantly higher BUN/creatinine ratio in the group with dynapenia/sarcopenia compared to the group without it. Creatinine production in the body relies on skeletal muscle mass, and a decrease in clearance due to renal impairment leads to an increase in serum concentration [41]. Furthermore, it has been reported that serum creatinine levels, when adjusted for renal function, positively correlate with skeletal muscle mass [42]. Conversely, BUN levels are also commonly used in clinical practice as an indicator of renal function; however, there are few studies addressing the relationship between BUN levels and skeletal muscle mass or sarcopenia. We demonstrated that the BUN/creatinine ratio relates to the diagnostic criteria for sarcopenia. Given its potential as an indicator of physical frailty, this ratio could be especially useful in monitoring muscle loss in schizophrenia patients, helping clinicians detect early signs of sarcopenia [3]. In fact, it has been shown that the BUN/creatinine ratio is associated with the physical frailty state that includes sarcopenia [17,18]. Additionally, the BUN/creatinine ratio can also increase due to dehydration and gastrointestinal bleeding [43,44,45]. However, the cases examined in this study did not include instances of poor dietary intake or gastrointestinal disorders that could cause dehydration.

When the BUN/creatinine ratio is greater than 20, the body is believed to lack carbohydrates and fats, resulting in insufficient energy and the use of protein for energy metabolism, a process known as protein catabolism [52]. It is crucial to ensure adequate protein and calorie intake to prevent sarcopenia and protein-energy wasting, which can lead to frailty [53]. Furthermore, studies have indicated that the BUN/creatinine ratio correlates with skeletal muscle index (SMI) and grip strength [17,18]. Thus, the BUN/creatinine ratio may also assist in screening for sarcopenia or frailty in chronically hospitalized patients with schizophrenia. In clinical practice, some of these conditions are more urgent than sarcopenia. The BUN/creatinine ratio can vary for various reasons but may serve as one of the indicators in the screening for dynapenia or sarcopenia.

Among patients with BI level 1 (no disability), none had a BUN/creatinine ratio of 10 or greater. All patients with BI levels 2–4 had a BUN/creatinine ratio of 10 or greater. It is noteworthy that patients with BI levels 1–2 were also found to have dynapenia/sarcopenia, which was thought to be a pathological condition in schizophrenia patients. There was an association between BI score and dynapenia/sarcopenia. Particularly with regard to ADL, it seems necessary to pay attention to muscle weakness in partially independent patients who score 60 points or more.

Individuals with major depressive disorder (MDD) are more likely to have decreased skeletal muscle mass, and loss of muscle strength rather than muscle mass has been reported to play a role in the increased risk of MDD [54]. A number of exacerbations of bipolar affective disorder (BAD) associated with sarcopenia may indicate increased sedentary behavior [55]. Sarcopenia is more common in patients with BAD than in the general population [55]. Therefore, dynapenia, sarcopenia, and malnutritional status should be a focus when treating patients with MDD, BAD, and schizophrenia.

### 4.3. Limitations

These findings highlight the importance of monitoring EPSs, conducting regular blood tests, and assessing body composition to detect malnutrition. However, given the small sample size in this study, which may lead to false negative results, a larger population is needed to further identify risk factors. Moreover, we did not examine inter-rater reliability. Additionally, while we measured blood parameters such as albumin and fasting blood glucose, the assessment of nutritional status was only partial. Malnutrition in schizophrenia often involves micronutrient deficiencies and dietary intake patterns, which were not addressed. Future research should focus on developing simple clinical indicators and conducting longitudinal studies with larger cohorts to better understand these associations.

## 5. Conclusions

This study investigated the relationship between chronic schizophrenia, EPSs, nutritional status, body composition, dynapenia, and sarcopenia. Among the participants, 32.4% had no loss of muscle mass or function, 39.7% had dynapenia, and 27.9% had sarcopenia. Significant correlations were found between EPSs (gait and muscle rigidity), SMI, grip strength, walking speed, and chlorpromazine equivalent dose. Negative symptoms were also significantly related to some EPSs. Blood tests revealed that dynapenia and sarcopenia were associated with low total protein, albumin, and fasting blood glucose levels, indicating poor nutrition. The BUN/creatinine ratio was significantly related to various diagnostic criteria for sarcopenia, including BMI, SMI, grip strength, walking speed, body fat, visceral fat, body water content, and bone mass. The ratio was higher in the dynapenia/sarcopenia group compared to those without these conditions. EPSs, BMI, grip strength, total protein, and albumin were identified as useful indicators for detecting the risk of dynapenia/sarcopenia in routine psychiatric care. There was an association between BI score and dynapenia/sarcopenia. Particularly with regard to ADL, it seems necessary to pay attention to muscle weakness in partially independent patients who score 60 points or more. The use of a body composition analyzer could enhance the accuracy of detecting dynapenia/sarcopenia. Finally, although the BUN/creatinine ratio can fluctuate due to various factors, such as dehydration and severe infection, it may still serve as a useful screening indicator for diagnosing dynapenia/sarcopenia.

## Figures and Tables

**Figure 1 healthcare-13-00048-f001:**
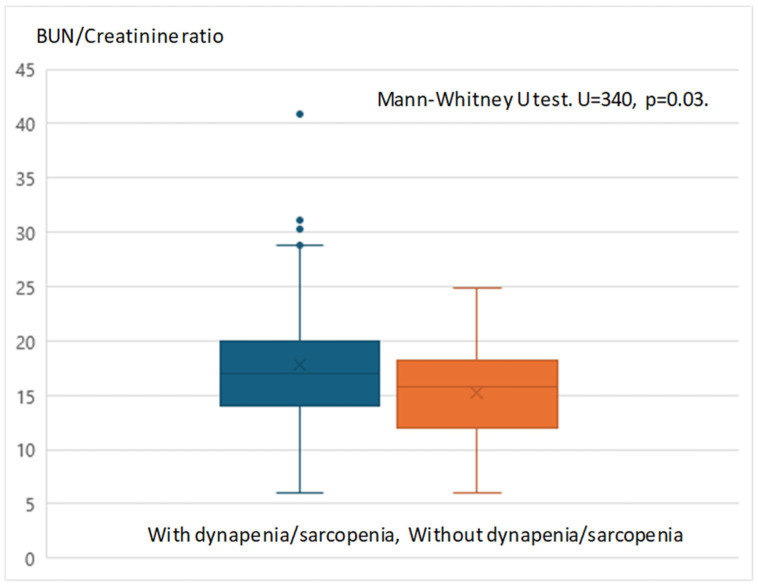
Comparison results between the two specific groups with and without dynapenia/sarcopenia. With dynapenia/sarcopenia (N = 46) vs. without dynapenia/sarcopenia (N = 22).

**Table 1 healthcare-13-00048-t001:** Descriptive data of all variables (N = 68).

Variables	Median (Min–Max)/ Frequency (%)
Demographics	
Age, years	64.0 (28–86)
Gender	
Male	34 (50.0)
Female	34 (50.0)
Clinical assessment	
DIEPSS	
Overall severity by total score (severity range: 0–4)	1.0 (0–3)
Overall severity by class	
None, normal	21 (30.9)
Very mild, uncertain	18 (26.5)
Mild	26 (38.2)
Moderate	3 (4.4)
Chlorpromazine equivalent dose, mg	629 (12.6–2382)
Global Assessment of Functioning (range: 0–100)	25 (5–35)
Brief Negative Symptoms Scale (range: 0–6)	2.9 (0.2–6.5)
Barthel Index (range: 0–100)	95.0 (15–100)
Muscle mass and function	
Classification	
Without dynapenia/sarcopenia	22 (32.4)
Dynapenia	27 (39.7)
Sarcopenia	19 (27.9)
Hand grip strength	
Left hand, kg	16.7 (6.0–41.0)
Right hand, kg	17.8 (6.0–40.7)
Muscle mass by skeletal muscle index (SMI)	6.525 (4.8–8.8)
Physical performance by walking speed, m/s	0.68 (0.02–1.3)
Body composition	
Body fat, %	29.1 (5.0–51.0)
Muscle mass, kg	37.25 (27.6–56.5)
Bone minerals, kg	2.2 (1.4–3.1)
Visceral fat level	8 (1–22)
Basal metabolic rate	1119.0 (12.3–1715.0)
Total body water, %	49.1 (36.7–76.0)
Height, cm	160 (139–177)
Weight, kg	56.5 (31.0–94.9)
Body mass index (BMI), kg/m^2^	22.4 (13.4–34.1)
Blood parameters	
Alanine aminotransferase (ALT), U/L	15 (5–89)
Aspartate aminotransferase (AST), U/L	18 (8–93)
Albumin (Alb), g/dL	3.9 (2.4–4.5)
Total cholesterol (TC), mg/dL	187 (98–261)
Triglycerides (TG), mg/dL	83.5 (33–342)
Gamma-glutamyl transferase (γ-GT), U/L	15.5 (6–166)
High-density lipoprotein cholesterol (HDL-C), mg/dL	52.2 (24.1–96)
Low-density lipoprotein cholesterol (LDL-C), mg/dL	115 (40–178)
Blood urea nitrogen (BUN), mg/dL	12.5 (5.2–66.8)
Creatinine, mg/dL	0.74 (0.4–3.3)
BUN/creatinine ratio	16.94 (6–40.9)
Blood sugar level	86 (68–114)
Uric acid (UA), mg/dL	4.4 (0.9–9.6)
Hemoglobin A1C (HbA1C), %	5.3 (4.4–6.7)

**Table 2 healthcare-13-00048-t002:** Spearman’s rho (ρ) correlation coefficients among sarcopenia, extrapyramidal symptoms, and negative symptoms (N = 68).

Variables	1	2	3	4	5	6	7	8	9	10	11	12	13	14	15	16	17	18
1. Age (Yrs)	—																	
2. Left Grip strength	−0.4 **	—																
3. Right Grip strength	−0.4 **	0.9 ***	—															
4. SMI	−0.3 *	0.7 ***	0.7 ***	—														
5. Walking speed (m/seconds)	−0.6 ***	0.3 *	0.3 *	0.3 *	—													
6. Chlorpromazine equivalent dose	−0.5 ***	0.2	0.2	0.1	0.3 *	—												
7. DIEPSS 1 Gait	0.3 *	−0.4 **	−0.4 **	−0.3 *	−0.5 ***	−0.3 *	—											
8. DIEPSS 2 Bradykinesia	0.4 **	−0.2 *	−0.3 *	−0.2	−0.4 ***	−0.4 ***	0.7 ***	—										
9. DIEPSS 3 Sialorrhea	0.1	−0.2	−0.2	−0.2	−0.2	−0.1	0.2	0.2	—									
10. DIEPSS 4 Muscle rigidity	0.2	−0.1	−0.1	−0.1	−0.1	−0.5 ***	0.3 *	0.5 ***	0.2	—								
11. DIEPSS 5 Tremor	0.2	−0.2	−0.2	−0.2	−0.3 *	−0.1	0.3 **	0.3 *	0.3 *	0.1	—							
12. DIEPSS 6 Akathisia	0.0	−0.0	−0.0	−0.0	−0.1	−0.1	0.3 *	0.3 *	0.3 **	0.2	0.0	—						
13. DIEPSS 7 Dystonia	−0.1	−0.1	0.0	−0.0	−0.1	−0.0	0.2	0.2	0.4 ***	0.1	0.1	0.4 ***	—					
14. DIEPSS 8 Dyskinesia	0.3 *	−0.0	−0.1	−0.1	−0.2	−0.2	0.2	0.2	0.4 ***	0.2	0.2	0.1	0.4 ***	—				
15. DIEPSS 9 Overall severity	0.3 *	−0.4 **	−0.3 **	−0.3 *	−0.3 *	−0.4 ***	0.7 ***	0.7 ***	0.4 ***	0.4 ***	0.4 ***	0.2	0.3 *	0.3 *	—			
16. BNSS total score	0.1	0.1	0.1	0.0	−0.1	−0.2	0.2	0.4 ***	−0.0	0.3 **	0.1	−0.1	−0.1	0.1	0.2	—		
17. F1 Motivational deficit total	0.1	0.1	0.1	0.0	−0.0	−0.2	0.2	0.4 ***	0.0	0.3 *	0.2	−0.	−0.1	0.1	0.3 *	0.9 ***	—	
18. F2 Expressive deficit total	0.1	0.2	0.1	−0.0	−0.1	−0.2	0.2	0.4 ***	−0.1	0.3 **	0.1	−0.2	−0.0	0.1	0.1	0.9 ***	0.7 ***	—

Note. * *p* < 0.05, ** *p* < 0.01, *** *p* < 0.001; Brief Negative Symptoms Scale (BNSS total) score; skeletal muscle index (SMI).

**Table 3 healthcare-13-00048-t003:** Correlation matrix on BUN/creatinine ratio and dynapenia/sarcopenia risk indicators (N = 68).

	1	2	3	4	5	6	7	8	9	10	11	12	13
1. BUN/creatinine ratio	—												
2. Age (Yrs)	0.4	—											
3. BMI (kg/m^2^)	−0.5 ***	−0.5 ***	—										
4. Body fat (%)	−0.3 **	−0.4 ***	0.8	—									
5. Visceral-Fat level	−0.5 ***	−0.4 ***	0.9	0.6	—								
6. Basal Metabolic Rate (kcal)	−0.5 ***	−0.5 ***	0.5	0.1	0.6	—							
7. Bone minerals (kg)	−0.5 ***	−0.4 ***	0.4	−0.0	0.6	0.9	—						
8. Body water (%)	0.4	0.5	−0.8 ***	−0.9 ***	−0.7 ***	−0.3 *	−0.1	—					
9. Trunk muscle mass (kg)	−0.5 ***	−0.3 **	0.4	−0.1	0.6	0.8	0.9	−0.1	—				
10. Left Grip strength (kg)	−0.3 *	−0.4 ***	0.3	−0.1	0.4	0.7	0.7	−0.0	0.7	—			
11. Right Grip strength (kg)	−0.3 **	−0.4 ***	0.3	−0.1	0.4	0.7	0.7	−0.1	0.7	0.9	—		
12. SMI	−0.4 ***	−0.3 **	0.5	0.0	0.5	0.8	0.9	−0.0	0.8	0.7	0.7	—	
13. Walking speed (m/seconds)	−0.3 **	−0.6 ***	0.2	0.2	0.2	0.3	0.4	−0.3	0.3	0.3	0.3	0.3	—

Note. Hₐ is negative correlation. Note. * *p* < 0.05, ** *p* < 0.01, *** *p* < 0.001. Spearman’s rho (ρ) correlation coefficients; body mass index (BMI); blood urea nitrogen/creatinine ratio (BUN/Cr Ratio); skeletal muscle index (SMI).

**Table 4 healthcare-13-00048-t004:** Comparison results of variables between groups defined by muscle mass and function: without dynapenia/sarcopenia vs. dynapenia vs. sarcopenia.

	Without Dynapenia/Sarcopenia (n = 22)	Dynapenia (n = 27)	Sarcopenia (n = 19)	Statistics	
	Median (Min–Max)	Median (Min–Max)	Median (Min–Max)	χ^2^	*p*-Value
Age (Yrs)	58 (28–73)	65 (35–86) a	73 (43–81) b	11.2	0.004
DIEPSS overall severity	0.5 (0–2)	2 (0–3)	2 (0–3)	5.6	0.06
CP equivalent dose (mg/day)	775 (100–2382)	600 (75–1300)	500 (12.6–1427)	5.2	0.075
BNSS total score	40 (8–65)	35 (8–84)	47 (2–81) a	8.9	0.012
Barthel Index	100 (65–100)	90 (30–100) b	85 (15–100) c	15.2	<0.001
Handgrip strength test, left (kg)	27.9 (18–41)	15.2 (6–25.8) c	13.8 (6–20.6) c	33.6	<0.001
Handgrip strength test, right (kg)	28.1 (18–40.7)	16.1 (10.2–25.5) c	14.1 (6–21.7) c	34.0	<0.001
Muscle mass (kg)	44.3 (32.3–56.5)	36.6 (28.4–48.1) b	33.4 (27.6–40.5) c	18.4	<0.001
Muscle mass left arm (kg)	2.25 (1.3–3.1)	1.65 (1.15–2.7) b	1.55 (1.1–3.05) b	14.5	<0.001
Muscle mass right arm (kg)	2.2 (1.3–6.6)	1.65 (1.2–3.0) a	1.4 (1.15–2) c	19.6	<0.001
Muscle mass left leg (kg)	7.22 (5.5–9.9)	6.4 (3.6–8.7) a	5.6 (4.1–7.7) b	13.8	0.001
Muscle mass right leg (kg)	7.47 (5.5–9.25)	6.4 (3.7–8.7) a	5.8 (3.6–7.4) c	19.5	<0.001
Trunk muscle mass (kg)	23.9 (18.5–31.8)	19.8 (14.5–26.4) a	19.4 (16.4–23.6) b	13.6	0.001
Skeletal muscle mass index (SMI)	7.1 (5.8–8.8)	6.5 (5.83–8.38) a	5.83 (4.83–6.73) c	30.3	<0.001
BMI (kg/m^2^)	23.9 (17.9–34.1)	22.9 (13.4–32.6)	18.8 (13.9–26.3) c	17.7	<0.001
Basal metabolic rate (kcal)	1331 (1015–1715)	1117 (1230–1385) b	999 (780–1131) c	27.7	<0.001
Bone mass (estimated weight of bone mineral), (kg)	2.45 (1.8–3.1)	2.2 (1.5–2.6) b	1.9 (1.4–2.3) c	23.3	<0.001
Body water (%)	48.5 (36.5–55.9)	49.1 (39–67)	49.1 (40–67)	2.8	0.25
Alanine aminotransferase (ALT) (U/L)	18.5 (7–41)	12 (5–89)	11 (5–36) a	6.8	0.033
Aspartate aminotransferase (AST) (U/L)	19.5 (12–32)	17 (8–93)	16 (13–28)	2.8	0.249
Gamma-glutamyl transferase (γ-GT) (U/L)	18.5 (9–115)	18 (9–96)	12 (6–166)	4.2	0.123
Total protein (TP) (g/dL)	7.1 (6–7.8)	7 (5.8–8) a	6.5 (5.9–8) b	12	0.002
Albumin (Alb) (g/dL)	4 (3.4–4.5)	4 (2.4–4.5)	3.8 (3–4.1) a	10.4	0.005
Albumin/Globulin Ratio (A/G Ratio)	1.35 (1.06–1.8)	1.32 (0.63–1.87)	1.22 (0.8–2.1)	0.6	0.739
Total cholesterol (T-Cho) (mg/dL)	187 (150–238)	192 (98–245)	186 (115–261)	0.2	0.913
High-density lipoprotein cholesterol (HDL-C) (mg/dL)	48 (31–80.6)	53.5 (24.1–96)	55.6 (40–84.6)	5.4	0.066
Low-density lipoprotein cholesterol (LDL-C) (mg/dL)	127 (68–176)	114 (50–171)	101 (40–178)	5.0	0.8
HDL/LDL Ratio	0.38 (0.2–1.2)	0.54 (0.16–1.21) a	0.504 (0.3–1.2) a	8.3	0.016
Triglycerides (TG) (mg/dL)	106 (49–215)	77 (44–342)	64 (33–258)	2.6	0.278
Fast blood sugar (FBS) (mg/dL)	86.5 (68–111)	87 (68–114) b	80 (71–92) b	15.5	<0.001
Hemoglobin A1c (HbA1c)	5.2 (4.5–5.8)	5.4 (4.4–6.7)	5.2 (4.8–6.0)	5.6	0.06
Blood urea nitrogen (BUN)	11.4 (5.2–14.8)	12.7 (5.3–21.8)	14.3 (10.1–66.8) b	8.8	0.013
Creatinine (mg/dL)	0.75 (0.5–1.0)	0.7 (0.44–1.11)	0.79 (0.35–3.34)	1.8	0.406
BUN/creatinine ratio	15.7 (5.98–24.9)	16.8 (10.7–40.9)	18.7 (13.3–31.1)	5.9	0.051
Body fat (%)	31.1 (19.4–47.1)	29 (5–51)	26.9 (5–45.4)	4.1	0.131
Visceral-fat level	11 (2.5–22)	9 (1–15.5)	5.5 (1–12.5) b	11.9	0.003
Uric acid (UA) (mg/dL)	4.45 (0.9–8.1)	4.5 (1.1–9.6)	3.5 (1.6–6)	2.3	0.316

Dwass–Steel–Critchlow–Fligner (DSCF) post hoc analysis. The reference point was set at the value observed in the without dynapenia/sarcopenia group. a: *p* < 0.05, b: *p* < 0.01, c: *p* < 0.001.

**Table 5 healthcare-13-00048-t005:** Association between BI level, dynapenia/sarcopenia, and BUN/creatinine ratio.

		Barthel Index Level				
Without-Sarcopenia/Dynapania/Sarcopenia		1 No Disability (91–100)	2 Mild (61–90)	3 Moderte (21–60)	4 Severe (0–20)	Total
Without	Observed	20	2	0	0	22
	% of total	29.4 %	2.9 %	0.0 %	0.0 %	32.4 %
Dynapenia	Observed	13	9	5	0	27
	% of total	19.1 %	13.2 %	7.4 %	0.0 %	39.7 %
Sarcopenia	Observed	5	9	3	2	19
	% of total	7.4 %	13.2 %	4.4 %	2.9 %	27.9 %
Total	Observed	38	20	8	2	68
Cramer’s V = 0.41	% of total	55.9 %	29.4 %	11.8 %	2.9 %	100.0 %

Gray cells indicate that patients have a BUN/creatinine ratio of 10 or greater. All patients with BI levels 2–4 had a BUN/creatinine ratio of 10 or greater.

**Table 6 healthcare-13-00048-t006:** Summary of comparable studies in the literature.

✓ Muscle weakness and drug-induced pyramidal symptoms contribute to impaired mobility [35,36]. ✓ Grip strength has been proposed as a biomarker for overall health status [37]. ✓ Negative symptoms and a tendency toward sedentary behaviors as factors [38], aging reduces muscle mass and is associated with sarcopenia [39,40]. ✓ Creatinine production in the body relies on skeletal muscle mass, and a decrease in clearance due to renal impairment leads to an increase in serum concentration [41]. ✓ Serum creatinine levels, when adjusted for renal function, positively correlate with skeletal muscle mass [42]. ✓ BUN/creatinine ratio is associated with the physical frailty state that includes sarcopenia [17,18]. ✓ BUN/creatinine ratio can also increase due to dehydration and gastrointestinal bleeding [43,44,45]. ✓ Schizophrenia often involves more severe physical decline, driven by disease progression and medication-induced motor dysfunction [46].

## Data Availability

The data presented in this study are available upon request from the corresponding author. The data are not publicly available due to privacy restrictions.

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
