# Peer review of "Association Between Dynapenia/Sarcopenia, Extrapyramidal Symptoms, Negative Symptoms, Body Composition, and Nutritional Status in Patients with Chronic Schizophrenia"

_healthcare, 2024, doi:10.3390/healthcare13010048_

Round 1

Reviewer 1 Report

Comments and Suggestions for Authors

In Table 1: a) Why you didn't show Stature, Body mass and BMI of subjects ? b) a bone mass of 2,2 kg (1,4-3,1) is IMPOSSIBLE !!!

Why didn't you used a better BIA device* to estimate body composition. You should use a multifrequency device able to estimate intra and extracellular water*.

(*) Some models of TANITA have this capabilities.

Author Response

Reviewer 1

We sincerely appreciate the reviewer’s kind consideration of our manuscript. The editor and reviewers’ constructive comments and invaluable suggestions have greatly helped improve the manuscript quality. In response to the suggestions, we have made comprehensive revisions. Details of these modifications are outlined in the resubmitted version. Additionally, we have provided a point-by-point response to each comment below, with our replies highlighted in yellow. We look forward to the reviewer’s favorable reply.

In Table 1:

  1. Why you didn't show Stature, Body mass and BMI of subjects ?

Response: We appreciate the reviewer’s detailed comments. We realize that we only listed the BMI in Table 1 on page 7. We have now added height and weight data for enhanced clarity.

  1. b) a bone mass of 2,2 kg (1,4-3,1) is IMPOSSIBLE !!!

Why didn't you used a better BIA device* to estimate body composition.

You should use a multifrequency device able to estimate intra and extracellular water*.

(*) Some models of TANITA have this capabilities.

Response: We thank the reviewer for this observation, that made us realize that we failed to include a definition of bone mass in the manuscript. We have amended this shortcoming by including the following definition:

Bone mass is the "estimated weight of bone mineral in the body" ( “2.5.4. Muscle mass and function”, page 5; Table 1, page 7; Table 3, page 11; and Table 4, page 12).

We also sincerely appreciate the multifrequency device recommendation. Unfortunately, limitations in research funding prevented us from accessing a multifrequency bioelectrical impedance analysis device to estimate intra and extracellular water. The model RD-545 InnerScan Pro (TANITA Corporation, Tokyo, Japan) we used to be appropriately priced for body composition measurement in psychiatric clinical practice without imposing a financial burden. We report the reference measurements here below.

Please note that the following template is provided as a reference for Asian people only.

https://www.tanita.asia/?_page=understanding&_lang=en&_para%5B0%5D=9&srsltid=AfmBOoqgOrF

Reviewer 2 Report

Comments and Suggestions for Authors

there are six areas where the manuscript can be improved to enhance its clarity, scientific rigor, and impact.

#1 introduction should be streamlined for ideas and avoid basic definitions e.g. schizophrenia

#2 problem statement need to be improved its currently unknown why this study was performed

#3 line 103 authors need 170 patients were enrolled, only 68 completed the study. this high attrition rate raises concerns about potential selection bias and the generalizability of the results. the inclusion and exlusion criteria are blurry its not clear if sample size of 68 is sfficuent this need to include power clulations

#4 blood parameters such as albumin and fasting blood glucose were measured, the assessment of nutritional status appears limited. malnutrition in schizophrenia often involves micronutrient deficiencies and dietary intake patterns, which were not addressed.

#5 authors in section 2.4 provided listing of parameters/variables without giving details on procedures e.g. for blood how many ml and where analysis done and what equipment used. for clinical assessment who did this? psychiatrist, research assistant etc.

specify the qualifications of the individuals conducting the assessments (e.g., psychiatrists, nurses, or research assistants) and whether they received standardized training for administering tools like diepss and bnss.

additionally, clarify if the assessors were blinded to participants' classification to minimize bias.

give procedural details for body composition analysis, including participant preparation (e.g., fasting requirements, hydration status) and measurement conditions (e.g., time of day, clothing worn during the procedure).

consider including a flow diagram to illustrate participant assessment.

#6 in discussion explain on how these findings can be translated into clinical practice, particularly for psychiatrists and dietitians managing patients with schizophrenia. how this different from bad or mdd??

Minor: all digits need to be scandalized to xx.x or xx.xx

Author Response

Reviewer 2

We sincerely appreciate the reviewer’s kind consideration of our manuscript. The editor and reviewers’ constructive comments and invaluable suggestions have greatly helped us improve the manuscript quality. In response to the suggestions, we have made comprehensive revisions. Details of these modifications are outlined in the resubmitted version. Additionally, we have provided a point-by-point response to each comment below, with our replies highlighted in yellow. We look forward to the reviewer’s favorable reply.

There are six areas where the manuscript can be improved to enhance its clarity, scientific rigor, and impact.

#1 introduction should be streamlined for ideas and avoid basic definitions e.g. schizophrenia

Response: We sincerely appreciate the reviewer’s advice and we have edited the Introduction accordingly. In particular, we have highlighted the significant impact of sarcopenia on daily life and the importance of clarifying the interrelation between risk indicators in patients with schizophrenia to help address invalidating symptoms (“Introduction”, page 2).

#2 problem statement need to be improved its currently unknown why this study was performed

Response: Thanks to the reviewer’s invaluable comment, we have completed the Introduction with a clear definition of the main aim and four sub-aims of this research. Specifically, the main purpose of the study is to determine the interrelation between dynapenia/sarcopenia, extrapyramidal symptoms, negative symptoms, body composition, and nutritional status in patients with schizophrenia (“Introduction”, page 2).

#3 line 103 authors need 170 patients were enrolled, only 68 completed the study. this high attrition rate raises concerns about potential selection bias and the generalizability of the results. the inclusion and exlusion criteria are blurry its not clear if sample size of 68 is sfficuent this need to include power clulations.

Response: We sincerely appreciate the reviewer's assessment of the manuscript. We have conducted the power analysis to determine the minimum sample size required for this study as detailed in the “2.6. Statistical methods” section on page 5. The inclusion and exclusion criteria have been further clarified in the “2.3.1. Inclusion criteria” and “2.3.2. Exclusion criteria” sections on page 3. Moreover, the section “3.1. Participants” on page 6 now reports the details of the 105 patients who were unable to undergo the complete set of measurements due to severe mental symptoms, and were therefore excluded from the analyses.

#4 blood parameters such as albumin and fasting blood glucose were measured, the assessment of nutritional status appears limited. malnutrition in schizophrenia often involves micronutrient deficiencies and dietary intake patterns, which were not addressed.

Response: Thanks to the reviewer’s insightful comment, we have added this limitation to the manuscript (“4.3. Limitations”, page 19).

#5 authors in section 2.4 provided listing of parameters/variables without giving details on procedures e.g. for blood how many ml and where analysis done and what equipment used.

Response: We thank the reviewer for identifying this omission. We have added the required information in the “2.5. 2. Blood Parameters” section on page 3 and the “2.5.4. Muscle mass and function” section on page 4.

for clinical assessment who did this? psychiatrist, research assistant etc.

specify the qualifications of the individuals conducting the assessments (e.g., psychiatrists, nurses, or research assistants) and whether they received standardized training for administering tools like diepss and bnss.

Response: We appreciate the reviewer’s detailed assessment of our manuscript and have now mentioned the qualification of the individuals conducting the assessments under “2.5.3. Clinical Assessment” on page 4.

additionally, clarify if the assessors were blinded to participants' classification to minimize bias.

Response: To minimize bias, the assessors were blinded to the participant's classification.

give procedural details for body composition analysis, including participant preparation (e.g., fasting requirements, hydration status) and measurement conditions (e.g., time of day, clothing worn during the procedure).

Response: We appreciate the reviewer’s request and have added the procedural details for body composition analysis, including participant preparation and measurement conditions, to section “2.5.4. Muscle mass and function” on page 4.

consider including a flow diagram to illustrate participant assessment.

Response: We appreciate the reviewer’s suggestion and believe we have now properly addressed this point in section “3.1. Participants” on page 6, by adding the detail of the patients whose datasets were excluded from the analyses.

#6 in discussion explain on how these findings can be translated into clinical practice, particularly for psychiatrists and dietitians managing patients with schizophrenia. how this different from bad or mdd??

Response: We greatly appreciate your invaluable comment. We agree that translating research findings into clinical practice is of utmost importance for the patients’ wellbeing. Therefore, we have added a distinct paragraph in the discussion section to interpret our results in light of these concepts, as suggested. Specifically, there was a causal relationship between sarcopenia-related traits and MDD and BAD. Also, we added two references.

Minor: all digits need to be scandalized to xx.x or xx.xx

Response: We appreciate the reviewer’s careful evaluation of our manuscript. We have rounded all digits to the first decimal place.

Reviewer 3 Report

Comments and Suggestions for Authors

The abstract needs to be revised to provide a clearer presentation of the study's objectives, methods, key findings, and conclusions. More emphasis should be placed on the intricate relationships between dynapenia, sarcopenia, extrapyramidal symptoms, negative symptoms, and body composition. The section must articulate the relevance of these associations to clinical care in schizophrenia. A dedicated subsection on limitations should be included, discussing sample constraints, methodology, and data limitations while proposing future research directions. To improve the reliability of the findings, additional validated assessment tools, such as the Short Physical Performance Battery (SPPB), Barthel Index, and Handgrip Strength Test, should be utilized. Justification for the sample size using G*Power analysis must also be provided, with the results detailed in the manuscript. The discussion section would benefit significantly from the inclusion of a table summarizing comparable studies from the literature, allowing for better contextualization of the findings. Finally, a table listing all abbreviations used in the manuscript should be added for clarity and ease of reference.  The study heavily relies on limited assessment tools; incorporating a broader range of validated tests is essential.

Absence of a G*Power analysis raises concerns about whether the study is adequately powered.

Inclusion criteria for participants are vague, especially regarding comorbidities, which could introduce sampling bias.

Inter-rater reliability measures, such as Cohen’s kappa, should be employed to ensure data consistency and enhance reproducibility.

Comments on the Quality of English Language

The English could be improved to more clearly express the research.

Author Response

Reviewer 3

We sincerely appreciate the reviewer’s kind consideration of our manuscript. The editor and reviewers’ constructive comments and invaluable suggestions have greatly helped us improve the manuscript quality. In response to the suggestions, we have made comprehensive revisions. Details of these modifications are outlined in the resubmitted version. Additionally, we have provided a point-by-point response to each comment below, with our replies highlighted in yellow. We look forward to the reviewer’s favorable reply.

  • The abstract needs to be revised to provide a clearer presentation of the study's objectives, methods, key findings, and conclusions.

Response: We are grateful to the reviewer for the thorough evaluation and insightful feedback. In response to the reviewer’s comment, we have edited the abstract to provide a clearer presentation of all the sections, with particular emphasis on the key findings of the present study.

More emphasis should be placed on the intricate relationships between dynapenia, sarcopenia, extrapyramidal symptoms, negative symptoms, and body composition. The section must articulate the relevance of these associations to clinical care in schizophrenia. A dedicated subsection on limitations should be included, discussing sample constraints, methodology, and data limitations while proposing future research directions.

Response: We are grateful to the reviewer for the reviewer’s recommendation. We have now added a “4.3. Limitations” section. 

  • To improve the reliability of the findings, additional validated assessment tools, such as the Short Physical Performance Battery (SPPB), Barthel Index, and Handgrip Strength Test, should be utilized.

Response: We appreciate the insightful comment regarding additional validated assessment tools. In response, we have included the Barthel Index data and Handgrip Strength test results in Table 4. Additionally, we have computed the relationship between Barthel Index level, Dynapenia/Sarcopenia, and BUN/Creatinine ratio to improve the reliability of the findings (Table 5).

  • Justification for the sample size using G*Power analysis must also be provided, with the results detailed in the manuscript. Inclusion criteria for participants are vague, especially regarding comorbidities, which could introduce sampling bias .

Response: We thank the reviewer for highlighting these potential drawbacks. We have edited the inclusion and exclusion criteria to improve clarity and specificity (“2.3.1. Inclusion criteria” and “2.3.2. Exclusion criteria”, page 3), and reported the G*Power analysis to justify the sample size (“2.6. Statistical methods”, page 5).

  • Inter-rater reliability measures, such as Cohen’s kappa, should be employed to ensure data consistency and enhance reproducibility.

Response: We thank the reviewer for bringing this to our attention. Unfortunately, inter-examiner reliability measures were not available for this investigation. This point has been listed as a limitation for the current study (“4.3. Limitations”, page 19). However, we agree with the reviewer about the importance of assessing inter-rater reliability and will incorporate the reviewer’s suggestion in future surveys.

  • The discussion section would benefit significantly from the inclusion of a table summarizing comparable studies from the literature, allowing for better contextualization of the findings.

Response: We appreciate this suggestion, as it adds context to the significance of our study’s findings. In response, we have included a table summarizing comparable studies from the literature (“4.2. Interpretation and Generalizability”, Table 6, page 17).

  • Finally, a table listing all abbreviations used in the manuscript should be added for clarity and ease of reference.

Response: Thanks to the reviewer’s feedback, we have listed all acronyms used in the manuscript in a dedicate section at the end of the main text (“Abbreviations”, page 20).  

  • The study heavily relies on limited assessment tools; incorporating a broader range of validated tests is essential. Absence of a G*Power analysis raises concerns about whether the study is adequately powered. Inclusion criteria for participants are vague, especially regarding comorbidities, which could introduce sampling bias.

Response: We sincerely appreciate the reviewer's insightful considerations. We have conducted the power analysis to determine the minimum sample size required for this study as detailed in the “2.6. Statistical methods” section on page 5. The inclusion and exclusion criteria have been further clarified in the “2.3.1. Inclusion criteria” and “2.3.2. Exclusion criteria” sections on page 3.

Round 2

Reviewer 1 Report

Comments and Suggestions for Authors

As mentioned in my previous mail, it's a shame that despite being a well methodologically presented study by different professionals from the biomedical field and with great potential scientific and practical interest, it continues to have such a significant error regarding the bone mass' estimation by a BIA system. The authors* should know that the estimation of bone's mineral content (not BONE MASS !) of any BIA system (even the most technologically advanced) has a very poor validity. In conclusion, in Table 1,3,4: The words/concept "Bone mass (estimates weight bone mineral)" have to be changed by just: "Bone minerals".  Also in lines 198-9: erase "bone mass and estimated" In line 200: where is written: "this device can perform estimate" the word "perform" has no meaning.

(*) Whether we like it or not, research in any scientific field requires specialists in each field of study. In this case, although I'm sure of the professionalism of this paper's authors in the field of Health sciences & Neuropsychiatry, I dare to recommend that for future studies consult a specialist in the field of body composition assessment. Luckily, you've one of best in Tokio. A Criterion Anthropometrist of the International Society for the Advancement of Kinanthropometry (ISAK), Dr. Masaharu KAGAWA (masaharuk@hotmail.com).

Author Response

Reviewer 1

We sincerely appreciate the reviewer's kind consideration of our manuscript. In response to the suggestions, we have made minor revisions. Details of these changes are highlighted in red in the resubmitted version. In addition, we have provided a point-by-point response to each comment below, with our responses highlighted in yellow.

As mentioned in my previous mail, it's a shame that despite being a well methodologically presented study by different professionals from the biomedical field and with great potential scientific and practical interest, it continues to have such a significant error regarding the bone mass' estimation by a BIA system. The authors* should know that the estimation of bone's mineral content (not BONE MASS !) of any BIA system (even the most technologically advanced) has a very poor validity. 

Response: Thank you for teaching us from a professional perspective.

In conclusion, in Table 1,3,4: The words/concept "Bone mass (estimates weight bone mineral)" have to be changed by just: "Bone minerals".  

Response: We have changed the words according to your advice. 

Also in lines 198-9: erase "bone mass and estimated" In line 200: where is written: "this device can perform estimate" the word "perform" has no meaning.

Response: We have deleted "bone mass and estimated" and removed "perform" and changed it to "estimate". Thank you very much.

(*) Whether we like it or not, research in any scientific field requires specialists in each field of study. In this case, although I'm sure of the professionalism of this paper's authors in the field of Health sciences & Neuropsychiatry, I dare to recommend that for future studies consult a specialist in the field of body composition assessment. Luckily, you've one of best in Tokio. A Criterion Anthropometrist of the International Society for the Advancement of Kinanthropometry (ISAK), Dr. Masaharu KAGAWA (masaharuk@hotmail.com).

Response: We would like to further develop this clinical research by consulting with Dr. Kagawa, a specialist in body composition assessment.

Once again, our heartfelt thanks for your kind advice on our manuscript.

Reviewer 2 Report

Comments and Suggestions for Authors

Thank you for addressing my concerns. 

Author Response

Comments and Suggestions for Authors: Thank you for addressing my concerns.

Response: We sincerely appreciate the reviewer’s kind consideration of our manuscript.

Reviewer 3 Report

Comments and Suggestions for Authors The authors have completely addressed all my comments, and I have no further concerns. Therefore, I recommend accepting the paper.

Author Response

Comments and Suggestions for Authors: The authors have completely addressed all my comments, and I have no further concerns. Therefore, I recommend accepting the paper.

Response: We sincerely appreciate the reviewer’s kind consideration of our manuscript.